

# Revisiting the role of vertical shear in analytic ice shelf models

Chris Miele[1], Timothy C. Bartholomaus[1], and Ellyn M. Enderlin[2]

[1]Department of Geological Sciences, University of Idaho, Moscow, ID, USA
[2]Department of Geosciences, Boise State University, Boise, ID, USA

**Correspondence:** Chris Miele (cmiele@uidaho.edu)

**Abstract.** Analytic modeling of ice shelf flow began when Weertman derived an expression for the strain rates within an unconfined shelf, of uniform thickness, extending only in one direction. Nearly two decades later, Thomas generalized Weertman's analysis to ice shelves of nonuniform thickness, deriving one of the most well-known analytic models in glaciology: $\overline{\tau_{xx}} = \frac{1}{4}\rho g h$. However, despite the prevalence of this model in both historical and contemporary texts, there remain persistent
miscommunications regarding the role of vertical shear in its construction. In Thomas' original approach, vertical shear stress was considered negligible in the stress balance; in a significant contrast, the same model is typically derived in contemporary texts by the neglect of basal resistance. These two approaches are not equivalent, and yet, it remains common to misinterpret vertical shear stress as typically neglected in current ice shelf modeling studies. This manuscript provides clarification on this pervasive misconception. We emphasize that vertical shear stress should not be interpreted as negligible in the construction of
general shallow shelf models. However, we also demonstrate that the vertical shear stress inherent in Thomas' expression does not give rise to a well-defined vertical shear strain rate. For situations in which vertical shear stress in shallow ice shelf models is of interest, we provide guidance on how to best calculate it.

## 1 Introduction

In modern glaciological models, the equations governing the flow of ice are typically solved using numerical methods. How-
ever, much of the field's history is based on analytic models – equations describing the behaviour of ice under constraints simplistic enough to permit solution by hand (Nye, 1951; Weertman, 1957; Reeh, 1968; Thomas, 1973a). While necessarily much more restricted in their application than general numerical methods of solution, the construction of analytic models permits quick back-of-the-envelope calculations, builds physical intuition, and, ultimately, guides the understanding upon which we build our more sophisticated numerical solvers. As such, analytic models remain strongly endorsed as a first object of
serious study for modern ice sheet modelers (Oerlemans, 2021).

In an analytic modeling context, floating ice shelves are of particular interest. Floating glaciers represent a unique opportunity to study ice dynamical processes under uncommonly simple constraints. Beneath grounded glaciers, for instance, conditions at the bed exert significant control on the flow of overlying ice. However, only sparse observations of the subglacial environment have been made, and our understanding of basal processes remains fundamentally limited (Stearns and van der Veen, 2019).
Underneath ice shelves, much of this ambiguity is removed: the ice floats over effectively inviscid seawater, so that the "bed" provides no resistance to flow, vastly simplifying the force balance and internal velocity structure. Moreover, with an ice shelf



in isostatic equilibrium with seawater, the thickness of a shelf can be reasonably well estimated by surface measurements alone. This is not the case for more general glacier settings, which require additional datasets for estimating thickness (Farinotti et al., 2009; Morlighem, 2017).

In large part due to the simplifications possible at ice shelves, these locations have been the preferred setting of many pioneering analytic modeling studies exploring ice dynamics and rheology (Weertman, 1957; Thomas, 1973a, b; Reeh, 1968; Sanderson, 1979). Analytic models taking advantage of these convenient ice shelf properties date back to at least 1957, when Weertman derived an expression for the tension and velocity gradients within a uniform-thickness ice shelf in uniaxial extension (that is, extension only in the longitudinal, or downstream, direction). Weertman found that, for a shelf with uniform

surface elevation $h = h_T$ (and, therefore, uniform thickness), the depth-averaged longitudinal deviatoric tension, $\overline{\tau_{xx}}$, could be calculated via

$$\overline{\tau_{xx}} = \frac{1}{4}\rho_i g h_T. \tag{1}$$

Further assuming that strain rates did not vary with depth, Weertman then used a depth-averaged constitutive relation to obtain an analytic solution for the velocity field within a shelf of uniform thickness.

Nearly two decades later, Thomas (1973a) set out to generalize Weertman's expression to shelves of nonuniform thickness. Using the same underlying assumptions as Weertman, but imposing no restrictions on the surface elevation $h$, Thomas obtained an expression nearly identical to Weertman's, wherein the depth-averaged deviatoric tension is

$$\overline{\tau_{xx}} = \frac{1}{4}\rho_i g h. \tag{2}$$

By Thomas' analysis, Weertman's solution is valid regardless of how $h$ varies along a shelf. Thomas' expression remains the
generally-accepted description of a nonuniform-thickness shelf in uniaxial extension, and it is routinely cited or independently derived in the literature (van der Veen, 1985; Sanderson, 1979; Cuffey and Paterson, 2010; Gudmundsson, 2013; Hughes, 2003; Oerlemans, 2021; Millstein et al., 2022). However, though Equation 2 has persisted, the formulation of this model has quietly undergone a conceptual shift over the decades. This conceptual shift relates to the role of vertical shear in ice shelves – a topic which, we argue, is associated with persistent miscommunications, and on which we seek to provide clarification.

In originally deriving his nonuniform-thickness model, Thomas' primary assumption was that vertical shear stress (the stress orientation associated with vertical gradients in horizontal velocity) was negligible in the stress balance. The neglect of vertical shear stress was universal in the formulation of ice shelf models at the time (Weertman, 1957; Thomas, 1973a; Robin, 1975; Sanderson, 1979). However, it was understood by some authors to be theoretically suspect. Sanderson and Doake (1979), for example, argued that vertical shear was fundamentally linked with the thickness gradient of an ice shelf, and that,

strictly speaking, vertical shear could not be zero except in the case of uniform thickness. This observation did not challenge the practical utility of Equation 2 (Sanderson and Doake (1979) found vertical shear to be small enough that its neglect was usually justified), but it highlighted a relationship that had been missed in Thomas' analysis.

The formulation of Thomas' model evolved with the development of the Shallow Shelf Approximation (SSA) (Morland, 1987; MacAyeal, 1989). The SSA, besides representing a leap forward in computational glaciology, was accompanied by a



subtle shift in the way ice shelf mechanics was conceptualized. Instead of neglecting vertical shear outright, the SSA omits basal resistance – a condition which, as we demonstrate in the main text, is consistent with nonzero vertical shear in the stress balance (Weis et al., 1999). Using the SSA as a starting point to derive an analytic model for an extending ice shelf, Thomas' Equation 2 results, but, this time, without the assumption of vanishing vertical shear. This is the modern approach to deriving Equation 2. However, the presence of vertical shear in the Thomas solution can be somewhat counterintuitive, resulting in persistent mischaracterizations of common practice, wherein many authors still interpret vertical shear as absent in contemporary ice shelf analysis,[1] while others explain that, although vertical shear is not *explicitly* retained in leading-order models, the term is, nonetheless, present at higher order (Christian Schoof, pers. comm., 2022).

With this manuscript, we provide conceptual guidance on the role of vertical shear in ice shelves, with a particular interest in the analytic model of Equation 2, both in its historical and modern formulations. By taking an analytic approach, we are able to explore the relationships between small quantities (i.e., vertical shear and thickness gradients). We clarify that vertical shear *stress* is not neglected from the stress balance in the modern construction of Equation 2. However, we also demonstrate that the vertical shear stress is fully decoupled from the velocity field, and, in that sense, cannot be used to calculate a vertical shear strain rate. This is possible because typical numerical use of Thomas' model does not require strict internal consistency between the flow field and the stress field. Such inconsistency commonly results from any approximate solution to the Stokes equations. We show that the vertical shear stress consistent with Equation 2 can only be incorporated into a velocity solution by eschewing the plug flow assumption. Strict internal consistency between the flow field and the stress field, which is typically not a goal of approximate, numerical solutions, requires that the base of a nonuniform-thickness ice shelf move more slowly than the surface.

This paper is laid out as follows. We begin by deriving, from first principles, the historical models of Weertman (1957) and Thomas (1973a). We demonstrate that Thomas' model, under the historical assumption of vanishing vertical shear, implicitly assumes Weertman's uniform-thickness property. This illustrates the necessary relationship, first discussed by Sanderson and Doake (1979), between vertical shear and thickness gradients in ice shelves. We then derive Equation 2 in its modern interpretation, via the SSA. While this approach does indeed produce Equation 2 as a solution to the stress balance permitting vertical shear, we demonstrate that, due to the SSA's assumption of plug flow, no analytic velocity field can be strictly consistent with this stress solution. As stated earlier, this is expected from approximate solutions; however, we find it worthwhile to demonstrate what an internally consistent velocity field looks like in a nonuniform-thickness ice shelf. To this end, we construct an analytic model for an isothermal ice ramp with linear rheology, and we compare our velocity solution with that which would arise from the Thomas model. The two agree exactly at the waterline, and diverge the most noticeably at the base.

---

[1]For example, in constructing the ice shelf model of Pattyn and Decleir (1995), "the [vertical] shear stress term in [the $x$-momentum equation] is omitted." Bueler and Brown (2009) state that "$D(\mathbf{v})_{13}, D(\mathbf{v})_{31}, D(\mathbf{v})_{23}, D(\mathbf{v})_{32}$ [i.e., the vertical shear strain rates] are all negligible in the SSA." Cuffey and Paterson (2010) specify that, to construct a nonuniform-thickness ice shelf model, the "assumption must be made that the slope at the bottom surface of the shelf is small so that the stress $\tau_{xz}$ will be negligible." Larour et al. (2012) introduce the SSA as obtained by "assuming that vertical shear is negligible," and then specify that $\dot{\varepsilon}_{xz} = \dot{\varepsilon}_{yz} = 0$. In an ice shelf model intercomparison, Pattyn et al. (2013) write, "A further approximation, known as the shallow-shelf approximation (SSA), is obtained by neglecting vertical shear." Bondzio et al. (2016) describe the SSA as an approximation which "neglects all vertical shearing but includes membrane stresses," and Rückamp et al. (2019) affirm that "the SSA neglects vertical shearing."





Our clarifications provide a cautionary note against the construction of shear-free ice shelf models. As we demonstrate in
the discussion, the neglect of vertical shear stress can result in ice shelf models which are both physically and numerically
implausible. Secondarily, for situations in which the magnitude of the vertical shear stress is of interest, we provide guidance
on how best to calculate it.

## 2    Conventions, preliminary derivations, and ice shelf assumptions

We adopt the coordinate system of both Weertman (1957) and Thomas (1973a), taking $z$ to be the true vertical, with $x$ the
horizontal direction closest corresponding to flow. Since our primary interest is in uniaxially extending ice shelf models, we
consider only the two-dimensional (2D) domain depicted in Figure 1, neglecting any lateral shear stresses or lateral flow (this
is a significant simplification; due to the large stress-coupling lengths typical of floating ice, the dynamics of shelves are easily
impacted by distant lateral obstructions, such as sidewalls and ice rises).

### 2.1    Fundamental governing equations

The ice shelf models discussed here are built on the $x$ and $z$ momentum equations, from which we neglect all lateral shear
terms. These equations are expressed

$$\frac{\partial}{\partial x}\sigma_{xx} + \frac{\partial}{\partial z}\sigma_{xz} = 0 \tag{3a}$$

$$\frac{\partial}{\partial x}\sigma_{zx} + \frac{\partial}{\partial z}\sigma_{zz} = \rho_i g, \tag{3b}$$

where $g$ is the gravitational constant and $\rho_i$ is the density of ice. The terms $\sigma_{ij}$ denote the net stress acting on the $i$ face in the $j$
direction; since the momentum equations describe a body in equilibrium, there can be no net torque, and so the stress tensor is
symmetric and $\sigma_{ij} = \sigma_{ji}$. Each net stress can be partitioned into the sum of the mean normal stress $\sigma_M = \frac{1}{3}(\sigma_{xx} + \sigma_{yy} + \sigma_{zz})$
and a deviatoric component $\tau_{ij}$, with

$$\sigma_{ij} = \tau_{ij} + \delta_{ij}\sigma_M, \tag{4}$$

where $\delta_{ij}$ is the Kronecker delta, which takes the value 1 when $i = j$ and 0 otherwise. By this definition, for shear stresses
$i \neq j$, $\sigma_{ij} = \tau_{ij}$. It follows from Equation 4 that, neglecting lateral extension, $\tau_{zz} = -\tau_{xx}$. Meanwhile, the horizontal gradient
in vertical shear stress, $\frac{\partial}{\partial x}\sigma_{zx}$, gives rise to what is often called the "bridging term" or "T-term," and this term is typically
neglected from analyses of glacier flow (van der Veen and Whillans, 1989; Greve and Blatter, 2009; Cuffey and Paterson,
2010) even when the vertical shear itself is non-negligible. Integrating Equation 3b from arbitrary $z$ to the surface elevation $h$,
under the neglect of the bridging term, provides the following description of the mean normal stress, $\sigma_M$.

$$\sigma_M = \rho_i g(z - h) + \tau_{xx} \tag{5}$$



Here, we have neglected atmospheric pressure so that $\sigma_{zz}|_{z=h} = 0$. We now partition the full stresses in Equation 3a via Equation 4, under the present expression for $\sigma_M$, to produce the fundamental governing equation

$$2\frac{\partial}{\partial x}\tau_{xx} + \frac{\partial}{\partial z}\tau_{xz} = \rho_i g \frac{\partial}{\partial x}h. \tag{6}$$

Although this governing equation contains components from both Equations 3a and 3b, for brevity, we refer to it as the $x$-momentum equation. It is this equation from which both the Weertman and Thomas models arise, given the simplifying ice shelf assumptions discussed below.

## 2.2 Simplifying ice shelf assumptions

The ice shelf models discussed in this manuscript are built on the following typical assumptions. First, we assume shelf ice to have spatially uniform density, so that $\rho_i$ is a constant (as is the density of the underlying seawater, $\rho_w$). While the uniform-density assumption likely overestimates the total mass of an ice column, it is possible to compensate for this by decreasing the effective thickness of a shelf to account for less-dense snow or firn. We also assume the shelf to be in perfect isostatic equilibrium with seawater. With $z = 0$ the waterline, this condition allows both the surface elevation $h$ and the basal elevation $b$ to be expressed in terms of the full thickness $H$, with

$$h = \left(1 - \frac{\rho_i}{\rho_w}\right)H \tag{7a}$$

$$b = -\frac{\rho_i}{\rho_w}H. \tag{7b}$$

Our next assumption is that horizontal velocities are independent of depth (this is the plug flow condition mentioned earlier). In our 2D setup, the only horizontal velocity component is $u_x$, the velocity in the $x$-direction, and this assumption can be succinctly expressed as the requirement that $\frac{\partial}{\partial z}u_x = 0$. This depth-invariant condition is universal in analytic ice shelf models, and it is a key simplifying assumption in the SSA (MacAyeal, 1989).

For the sake of reconstructing the original workflow of Weertman (1957) and Thomas (1973a), we temporarily introduce the historical condition that vertical shear stress $\tau_{xz}$, within a floating shelf, is zero. We will show that this assumption results in the uniform-thickness property when it is used to derive Equation 2, emphasizing the necessary relationship between vertical shear stress and thickness gradients in an ice shelf (Sanderson and Doake, 1979). Later, in the context of outlining the modern approach to the Thomas solution, via construction of the SSA, we will replace this assumption with the requirement that basal resistance at the bottom of the shelf is zero. In Weertman's study, which dealt only with uniform-thickness shelves (likely guided by the observation that surface elevation gradients in ice shelves are typically very small), the basal plane is horizontal, and the distinction above is unimportant. However, the isostatic property of Equation 7 results in basal elevation gradients an order of magnitude larger than surface elevation gradients, and so the basal plane of an ice shelf may noticeably diverge from the horizontal in natural settings.



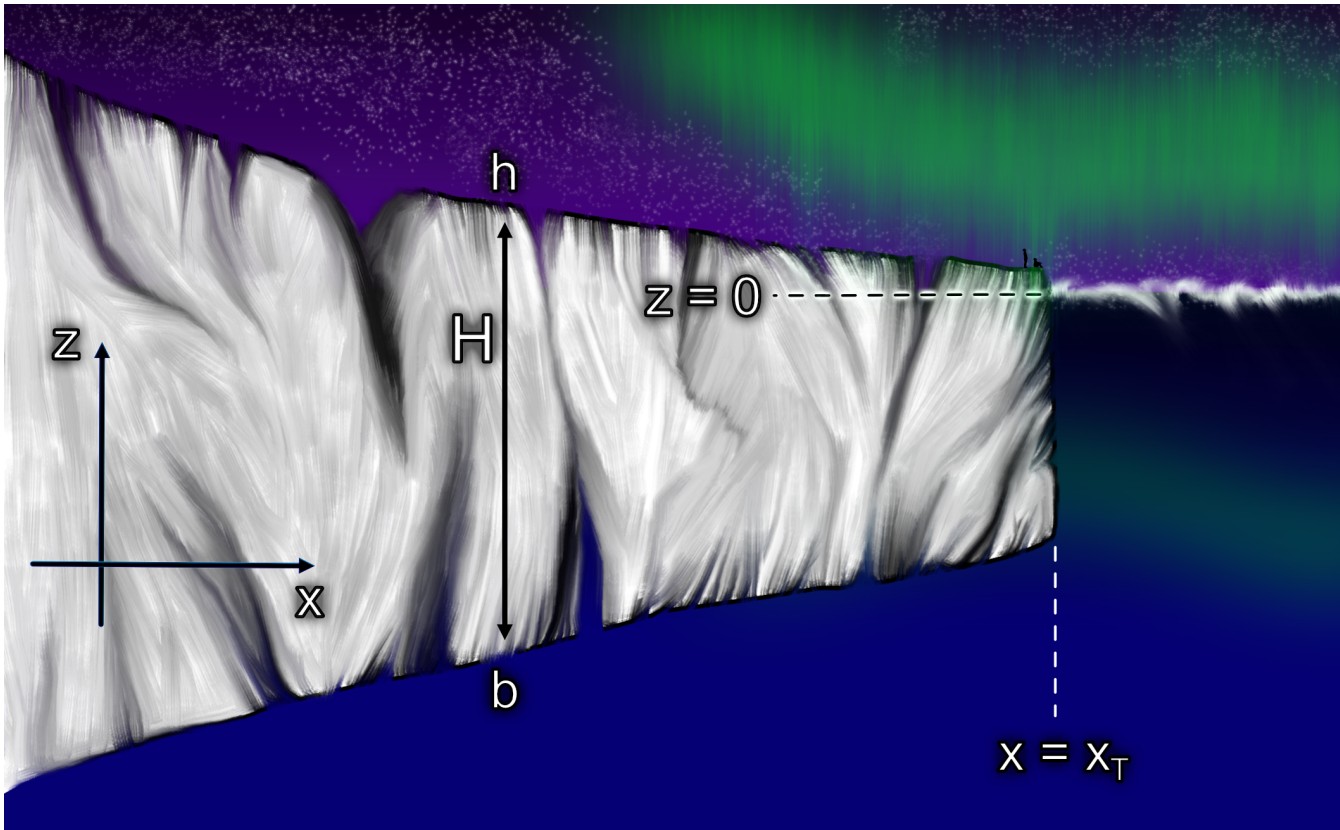

**Figure 1.** An ice shelf cross-section alongside a visual description of the geometric parameters relevant to this manuscript. $H$, $h$, and $b$ represent the thickness, surface elevation, and basal elevation of the shelf. $z = 0$ is the waterline and $x = x_T$ is the terminus. People on terminus for scale.

## 3 The uniform-thickness Weertman ice shelf

We construct the Weertman solution for a uniform-thickness shelf by applying the historical shear-free condition to Equation 6, while also restricting the surface elevation $h$ to uniformly equal its value at the terminus, $h_T$ (by isostacy, the basal profile is also level, resulting in a uniform-thickness shelf). For a uniform-thickness shelf, $\frac{\partial}{\partial x} h = 0$, and the result is the highly simplified

150    governing equation $\frac{\partial}{\partial x} \tau_{xx} = 0$.

In this case, $\tau_{xx}$ is a function of $z$ alone, and its depth-averaged value $\overline{\tau_{xx}}$ must everywhere be equal to the depth-averaged deviatoric stress due to the far-field effect of the terminal cliff. For any point of interest more than one or two ice thicknesses upstream of the terminus, a vertical cliff at terminal location $x = x_T$ induces a depth-averaged deviatoric stress $\overline{\tau_{xx}}(x_T)$ which emerges from the imbalance between the weight of the ice and the opposing weight of the seawater (Cuffey and Paterson,

155    2010). With the cliff at flotation, so that the terminal values of the surface elevation, basal elevation, and thickness ($h_T$, $b_T$, and $H_T$) relate to each other via Equation 7, it must hold that





$$\int\limits_{b_T}^{h_T} 2\tau_{xx}(x_T)dz = \int\limits_{b_T}^{h_T} \rho_i gzdz - \int\limits_{b_T}^{0} \rho_w gzdz, \tag{8}$$

where the first integral on the right represents the integrated ice overburden, and the second, the integrated seawater overburden (Cuffey and Paterson, 2010). The left-hand side is the depth-integrated longitudinal "resistive stress," which represents the difference between the full longitudinal stress and the ice overburden (van der Veen, 2013), and it can can be rewritten as $2H_T\overline{\tau_{xx}}(x_T)$. Evaluating the integrals on the right-hand side results in

$$2H_T\overline{\tau_{xx}}(x_T) = \frac{1}{2}\rho_i gH_T h_T. \tag{9}$$

Generalizations of Equation 9 are commonly applied as boundary conditions in ice sheet models terminating in floating cliffs (MacAyeal, 1989; Muszynski and Birchfield, 1987; Schoof, 2007; Goldberg et al., 2009). In constructing the Weertman model for a uniform-thickness ice shelf, the surface profile $h = h_T$ is uniform in $x$, and the depth-averaged value $\overline{\tau_{xx}}$ must uniformly be equal to $\overline{\tau_{xx}}(x_T)$ as induced by the distant ice cliff. Equation 1 emerges as an immediate result.

Equation 1, therefore, is Weertman's (1957) solution for the depth-averaged deviatoric stress within an unconfined shelf, with uniform thickness, under uniaxial extension. Under the previously-stated assumption that the horizontal velocity $u_x$ (and, therefore, the gradient $\frac{\partial}{\partial x}u_x$) does not vary with depth, Equation 1 can be used alongside a depth-averaged constitutive relation to determine the velocity field of a uniform-thickness shelf.

## 4 The historical Thomas ice shelf

Thomas (1973a) set out to generalize Equation 1 by requiring only that $\tau_{xz} = 0$, while imposing no restrictions on the shelf's thickness distribution. Although we will show this to be a contradiction, we outline his method below.

Partitioning the full stress $\sigma_{xx}$ into its deviatoric and mean normal components via Equations 4 and 5, the deviatoric stress $\tau_{xx}$ can be expressed

$$\tau_{xx} = \frac{1}{2}\left(\sigma_{xx} - \rho_i g(z - h)\right). \tag{10}$$

Depth-integrating Equation 10 over the full thickness of the shelf, we obtain

$$H\overline{\tau_{xx}} = \frac{1}{2}\left(\frac{1}{2}gH^2 - F\right), \tag{11}$$

where $F$ is defined as $F := -H\overline{\sigma_{xx}}$. Thomas' solution emerges from finding the appropriate value for $F$, which is interpreted as the total force pushing backward against a vertical section of shelf. This value is taken to be the force, per unit width, exerted horizontally against the shelf by the weight of the seawater. With this assumption, Thomas finds the value of $F$ to be





$$F = -\int_b^0 \rho_w gz \, dz = \frac{1}{2}\rho_i g \left(\frac{\rho_i}{\rho_w}\right) H^2, \tag{12}$$

where the isostatic condition of Equation 7 has been invoked to write $b$ in terms of $H$. Substituting Equation 12 into Equation 11, Thomas's unconfined ice shelf is governed by the Equation 2. Thus, Thomas' result provides consistency with Weertman's solution – if $h$ is uniform, Equation 1 results.

However, there is an odd detail regarding the Thomas model that emerges when we insert the stress solution of Equation 2 into the $x$-momentum equation. Thomas is forthright about his neglect of vertical shear (for example, refer to the abstract of Thomas (1973a)). In this case, as shown by Robin (1975), the governing equation, Equation 6, must reduce to

$$2\frac{\partial}{\partial x}\tau_{xx} = \rho_i g\frac{\partial}{\partial x}h. \tag{13}$$

Since no assumptions were made about the ice thermal structure in deriving Equation 2, there is no reason the model shouldn't hold for isothermal ice. In an isothermal shelf, $\tau_{xx}$ is equal to its depth-averaged value, with $\tau_{xx} = \overline{\tau_{xx}} = \frac{1}{4}\rho_i gh$. Inserting Thomas' isothermal description of $\tau_{xx}$ into Equation 13 demonstrates that the governing equation is apparently not satisfied by Thomas' own solution: the left-hand side evaluates to $\frac{1}{2}\rho_i g\frac{\partial}{\partial x}h$, rather than $\rho_i g\frac{\partial}{\partial x}h$. More specifically, for equality to hold, $\frac{\partial}{\partial x}h$ would have to be zero: despite setting out to construct a nonuniform-thickness shelf model, Thomas' isothermal solution satisfies his governing equation only with the uniform-thickness property. Below, we show that this result is not particular to the isothermal case.

### 4.1 Inconsistencies with Thomas' analysis

Here, we illustrate that, due to the neglect of vertical shear, Thomas' model is subject to the uniform-thickness property in the general, non-isothermal case. Let $\tau_{xx}^{\text{Thomas}}$ denote Thomas' solution for $\tau_{xx}$, in its non-depth-averaged sense, so that $\overline{\tau_{xx}^{\text{Thomas}}}$ is given by Equation 2. Regardless of whether it is assumed equal to its depth-averaged value, $\tau_{xx}^{\text{Thomas}}$ must satisfy the $x$-momentum equation. Under Thomas' condition that $\tau_{xz} = 0$, this is Equation 13. Integrating Equation 13 with respect to $x$, $\tau_{xx}^{\text{Thomas}}$ satisfies

$$\tau_{xx}^{\text{Thomas}} = \frac{1}{2}\rho_i gh + C(z), \tag{14}$$

for some "constant" of integration $C(z)$. Depth-averaging both sides then provides an alternative description of Thomas' depth-averaged solution, with

$$\overline{\tau_{xx}^{\text{Thomas}}} = \frac{1}{2}\rho_i gh + \overline{C(z)}. \tag{15}$$





$\overline{C(z)}$ is now a true constant. Using Equation 2 to rewrite the left-hand side, we find that

$$-\frac{1}{4}\rho_i gh = \overline{C(z)}. \tag{16}$$

That is, $-\frac{1}{4}\rho_i gh$ is equal to a constant, which requires that $h$ be uniform. Thomas' approach, though undertaken with the intention of generalizing the Weertman solution, is equivalent to the Weertman solution.

By pointing out Thomas' unintentional use of the uniform-thickness property, we in no way imply that Thomas' model is incapable of approximating the velocity fields of nonuniform-thickness ice shelves. It is typical to apply mathematical models to situations in which they do not apply exactly (Weertman (1957) himself applies his uniform-thickness model to a real-life setting at Maudheim, for example, despite the understanding that no real-life setting has perfectly uniform thickness). Rather, by drawing attention to the equivalence between the approaches of Thomas and Weertman, we emphasize the necessary relationship between vertical shear and thickness gradients in an ice shelf – to neglect one is to neglect the other.

## 5 The modern approach to the Thomas solution

In this section, we obtain Equation 2 by the modern approach, which neglects basal resistance without assuming vertical shear to be zero in the stress balance. Whereas Thomas' original approach began by depth-integrating an expression for the full stress $\sigma_{xx}$, the modern approach to constructing the Thomas solution requires, instead, the depth-integration of the $x$-momentum equation (MacAyeal, 1989; Morland, 1987; Weis et al., 1999). We outline this procedure below.

### 5.1 Depth-averaging the $x$-momentum equation

The following workflow closely follows van der Veen and Whillans (1989), although we continue to express the equations in terms of deviatoric stresses (Greve and Blatter, 2009; Cuffey and Paterson, 2010) rather than using the exact expressions of van der Veen and Whillans (1989).

After depth integrating the $x$-momentum equation (Equation 6) from the glacier base $b$ to surface $h$ and applying the Leibniz Rule to move derivatives outside integrals, we obtain

$$\left(2\frac{\partial}{\partial x}\left(H\overline{\tau_{xx}}\right) - 2\tau_{xx}|_{z=h}\frac{\partial}{\partial x}h + 2\tau_{xx}|_{z=b}\frac{\partial}{\partial x}b\right) + \left(\tau_{xz}|_{z=h} - \tau_{xz}|_{z=b}\right) = \rho_i gH\frac{\partial}{\partial x}h. \tag{17}$$

We reduce this expression by regrouping terms into naturally-arising definitions. For example, basal resistance results from shears at the bottom of a glacier acting parallel to the basal plane, which is generally not parallel to the horizontal $z$ plane. A coordinate transformation (Greve and Blatter, 2009) characterizes the $x$ component of the basal resistance, $\tau_{bx}$, as

$$\tau_{bx} = -2\tau_{xx}|_{z=b}\frac{\partial}{\partial x}b + \tau_{xz}|_{z=b}. \tag{18}$$





A similar definition holds for the $x$ component of any surface resistance, $\tau_{sx}$, with

$$\tau_{sx} = -2\tau_{xx}|_{z=h}\frac{\partial}{\partial x}h + \tau_{xz}|_{z=h}. \tag{19}$$

Any natural glacier setting has $\tau_{sx} = 0$ (van der Veen and Whillans, 1989). Substituting the definitions from Equations 18 and 19 into Equation 17, we find that

$$-2\frac{\partial}{\partial x}\left(H\overline{\tau_{xx}}\right) + \tau_{bx} - \tau_{sx} = -\rho_i gH\frac{\partial}{\partial x}h, \tag{20}$$

where the term on the right is the $x$ component of the driving stress, often symbolized $\tau_{dx}$. Equation 20 is the depth-integrated $x$-momentum equation, and it is equivalent to Equation 6, as no additional assumptions have been invoked in derivation.

**5.2     Obtaining the Thomas solution via the SSA**

Here, we demonstrate the standard construction of an analytic, uniaxial ice shelf model from Equation 20. As an ice shelf extends, its base and surface are opposed only by seawater and air. Neglecting the effects of ocean currents and wind on glacier flow, there should be no resistance at either the surface or the base of a freely-floating ice shelf. Mathematically, these conditions are satisfied by setting $\tau_{bx}$ and $\tau_{sx}$ to zero in Equations 18 and 19. The resulting differential equation is

$$-2\frac{\partial}{\partial x}\left(H\overline{\tau_{xx}}\right) = -\rho_i gH\frac{\partial}{\partial x}h. \tag{21}$$

Equation 21 is the governing equation of the SSA, in its simplest-case scenario of uniaxial extension. It can be verified, by direct substitution, that the solution to Equation 21 is the Thomas model, Equation 2.

Importantly, this construction of the Thomas model did not invoke the shear-free condition. To the contrary, setting $\tau_{sx} = \tau_{bx} = 0$ explicitly required $\tau_{xz}$ to be *nonzero* for nonuniform-thickness shelves: if $\tau_{xz}$ were zero in Equations 18 and 19,

thickness would automatically be uniform. This provides a concise mathematical description of the relationship, emphasized in the previous section, regarding the connection between thickness gradients and vertical shear stress in ice shelves. By this method of construction, therefore, Equation 2 represents a stress field in which the thickness gradient is appropriately balanced by vertical shear, even though that vertical shear term has symbolically disappeared in the depth-integration process.

A natural supposition, then, is that a vertical shear strain rate should be meaningfully present in the velocity solution corre-

sponding to Equation 2. However, as we show in the next section, this is not the case. We illustrate that, under the conventional plug-flow condition, the vertical shear strain rate accompanying Thomas' model is, in fact, ill-defined: two distinct methods of calculation yield two distinct results. By pointing this out, we emphasize that, while the vertical shear stress is not neglected in constructing the Thomas model, the actual calculation of the vertical shear stress must be done with care. In general, it cannot be calculated via the velocity field. As we later show, this is because the plug-flow condition is not strictly compatible with

nonuniform-thickness shelf geometry.





## 6 Exploring the vertical shear strain rate of the Thomas solution

### 6.1 Two characterizations of strain rates

One of the key motivations for the construction of models like Equation 2 is the description of velocity fields and strain rates within flowing ice, and one of the key challenges of solving the equations of glacier flow is ensuring compatibility between

two distinct descriptions of these strain rates. First, and by definition, a stain rate $\dot{\varepsilon}_{ij}$ is given by

$$\dot{\varepsilon}_{ij} = \frac{1}{2}\frac{\partial}{\partial i}u_j + \frac{1}{2}\frac{\partial}{\partial j}u_i, \tag{22}$$

where $i$ and $j$ may take any values in $\{x, y, z\}$. For example, if $u_x$ and $u_z$ represent the horizontal and vertical velocity components, $\dot{\varepsilon}_{xx} = \frac{\partial}{\partial x}u_x$, while $\dot{\varepsilon}_{xz} = \frac{1}{2}\frac{\partial}{\partial z}u_x + \frac{1}{2}\frac{\partial}{\partial x}u_z$.

Second, the strain rate $\dot{\varepsilon}_{ij}$ relates to the deviatoric stress $\tau_{ij}$ via an empirical constitutive relation. Thus, a complementary

characterization of strain rates is given by an equation of the form

$$\dot{\varepsilon}_{ij} = A\tau_E^{n-1}\tau_{ij}. \tag{23}$$

Equation 23 is a typical formulation of the constitutive relation for flowing ice, where $A$ is a function of temperature and other physical parameters (Goldsby and Kohlstedt, 2001), and the flow exponent $n$ is typically taken to be three (Cuffey and Paterson, 2010). The term $\tau_E$ is the effective stress, which relates to the second invariant of the deviatoric stress tensor.

There is no *a priori* connection between these two characterizations: the first relates strain rates to velocities but makes no mention of stresses; the second relates strain rates to stresses without any mention of velocities. The two characterizations, taken together, induce a complex constraint on the relationship between orthogonal velocity components: to permit an internally consistent velocity field, any pair of orthogonal velocity components must relate to one another in a way that produces the same shear strain rate by definition (via Equation 22) as emerges from the constitutive relation (Equation 23). To appropriately model

fluid flow, it is necessary to ensure that Equations 22 and 23 are simultaneously satisfied.

We show, below, that the analytic model of Equation 2, as derived via the SSA, produces a velocity field which cannot simultaneously satisfy Equations 22 and 23 for $\dot{\varepsilon}_{xz}$, except under the additional constraint of a uniform-thickness, shear-free shelf. This inconsistency is a result of the approximate nature of the SSA.

### 6.2 The ill-defined vertical shear strain rate of the Thomas solution

In this section, we demonstrate that, in order to permit agreement between the two strain rate characterizations discussed above, the basal value $\tau_{xz}|_{z=b}$ is zero under a plug flow regime. This can only be the case when the basal elevation gradient is zero, so that thickness is uniform, and vertical shear is uniformly zero. In other words, the vertical shear strain rate is ill-defined for nonuniform-thickness geometry.

In our 2D setup, incompressibility requires that $\dot{\varepsilon}_{zz} = -\dot{\varepsilon}_{xx}$, and so we have that



$$\frac{\partial}{\partial z} u_z = -\dot{\varepsilon}_{xx},$$
(24)

where we have expressed the vertical compression via its velocity gradient definition. Integrating both sides from $z = 0$ to arbitrary elevation $z$, and noting that $\dot{\varepsilon}_{xx} = \frac{\partial}{\partial x} u_x$ cannot vary with depth under the plug-flow condition,

$$u_z - u_z|_{z=0} = -z\dot{\varepsilon}_{xx}.$$
(25)

By isostacy (and leaving mass balance processes out of our analysis), ice above and below the waterline move in vertically opposite directions. Therefore, no vertical motion is permitted at the waterline,[2] and so $u_z|_{z=0} = 0$. Next, because the horizontal velocity is depth-invariant, the vertical shear strain rate reduces, by definition via Equation 22, to $\dot{\varepsilon}_{xz} = \frac{1}{2}\frac{\partial}{\partial x} u_z$. In particular, the basal value of the vertical shear strain rate must satisfy $\dot{\varepsilon}_{xz}|_{z=b} = \frac{1}{2}\frac{\partial}{\partial x} u_z|_{z=b}$. It then follows from Equation 25 that

$$\dot{\varepsilon}_{xz}|_{z=b} = -\frac{1}{2} b \frac{\partial}{\partial x} \dot{\varepsilon}_{xx}.$$
(26)

This is the basal value of the vertical shear strain rate obtained by definition, via Equation 22, under the typical SSA simplifying
assumptions.

We now construct an alternative description of $\dot{\varepsilon}_{xz}|_{z=b}$ by invoking the empirical relationship of Equation 23. The basal boundary condition is satisfied by setting $\tau_{bx} = 0$ in Equation 18. Regardless of how the effective stress is defined, multiplying Equation 18 by the basal value $A\tau_E^{n-1}|_{z=b}$ expresses the basal boundary condition in terms of strain rates, and, noting that $\dot{\varepsilon}_{xx}|_{z=b} = \dot{\varepsilon}_{xx}$ is depth-invariant, this produces

$$\dot{\varepsilon}_{xz}|_{z=b} = 2\dot{\varepsilon}_{xx}\frac{\partial}{\partial x} b.$$
(27)

To permit a well-defined vertical shear strain rate, the descriptions of $\dot{\varepsilon}_{xz}|_{z=b}$ provided by Equations 26 and 27 must agree, requiring that

$$2\dot{\varepsilon}_{xx}\frac{\partial}{\partial x} b = -\frac{1}{2} b \frac{\partial}{\partial x} \dot{\varepsilon}_{xx}.$$
(28)

Next, we consider the signs of each term in Equation 28. By our choice of coordinate system, $b \leq 0$, as the shelf's base
lies beneath the waterline. In a natural unconfined ice shelf setting, thickness decreases in the downstream direction,[3] so

---

[2]The claim that $u_z|_{z=0} = 0$ can be defended more rigorously by pointing out the equivalence between ice shelves and ice streams under the SSA. It can be shown that the SSA-type solution for an ice stream of surface elevation $h$, with a slippery bed at $z = 0$, is $\overline{\tau_{xx}} = \frac{1}{4}\rho_i g h$ (Cuffey and Paterson, 2010). An ice shelf with surface elevation $h$ is governed by the same equation under the SSA. The subaerial portion of the shelf can, therefore, be equivalently modeled as an ice stream flowing over a level, shear-free, impenetrable bed.

[3]If thickness *increases* downstream, $\frac{\partial}{\partial x} b \leq 0$ while $\frac{\partial}{\partial x}\dot{\varepsilon}_{xx} \geq 0$, and the bounds in Equation 29 are simply reversed.



that the basal elevation rises with $x$, and $\frac{\partial}{\partial x}b \geq 0$. Additionally, an unconfined shelf should not be in compression, and so $\dot{\varepsilon}_{xx} \geq 0$. Finally, with the shelf thinning in the downstream direction, the rate of extension should not be increasing with $x$; that is, $\frac{\partial}{\partial x}\dot{\varepsilon}_{xx} \leq 0$. (These constraints are typical of natural unconfined ice shelf settings (Cuffey and Paterson, 2010), and they emerge from the conventional SSA solution discussed in this section.) Inserting these constraints, we find that the left-hand side of Equation 28 is bounded below by zero, while the right-hand side is bounded above by zero, with

$$0 \leq 2\dot{\varepsilon}_{xx}\frac{\partial}{\partial x}b = -\frac{1}{2}b\frac{\partial}{\partial x}\dot{\varepsilon}_{xx} \leq 0. \tag{29}$$

Both sides of Equation 29 are equal to the basal value $\dot{\varepsilon}_{xz}|_{z=b}$, and, therefore, it follows that $\dot{\varepsilon}_{xz}|_{z=b}$ must be zero to permit a well-defined vertical shear strain rate. As $\dot{\varepsilon}_{xz}$ should attain its largest magnitude at the shelf base, it would follow that $\dot{\varepsilon}_{xz} = 0$ uniformly. That is, under the assumption of plug-flow, the vertical shear strain rate is only well-defined for uniform-thickness shelves.

Stated another way, if the we interpret the thickness gradient as nonzero, then Equations 26 and 27 represent two distinct values of vertical shear calculable from the Thomas solution's velocity field. Moreover, the values are not numerically similar to one another: they have opposite signs. Therefore, even though $\tau_{xz}$ is not neglected in the construction of Equation 2, it should not be assumed that this value relates, via the associated strain rate $\dot{\varepsilon}_{xz}$, to the flow field.

## 7 Discussion

### 7.1 The Budd ice shelf

A key point of this manuscript has been to emphasize that vertical shear should not be interpreted as absent from the stress balance of a nonuniform-thickness ice shelf – even when that vertical shear stress is not strictly consistent with the velocity field. In this section, we emphasize the importance of the vertical shear stress by constructing the problematic ice shelf model which would arise from its neglect.

As shown earlier, the neglect of vertical shear from the stress balance results in the governing equation of Equation 13 (Robin, 1975). Following the workflow of Section 4.1, the depth-integrated deviatoric tension arising from this governing equation is

$$\overline{\tau_{xx}} = \frac{1}{2}\rho_i gh + \overline{C(z)}, \tag{30}$$

where $\overline{C(z)}$, as a depth-averaged value, is constant. This expression for the depth-averaged stress is the one Thomas (1973a) associates with Budd (1969); consequently, we refer to Equation 30 as the Budd ice shelf. Using the conventional ice cliff boundary condition of Equation 9, the Budd ice shelf can be further simplified to

$$\overline{\tau_{xx}} = \frac{1}{2}\rho_i gh - \frac{1}{4}\rho_i gh_T. \tag{31}$$





It is almost trivial to construct the ice shelf model of Equation 31 if we (incorrectly) interpret vertical shear stress as negli-
gible. To see the error with this interpretation, notice that $\frac{1}{4}\rho_i g h_T \geq 0$, and so the second term acts to decrease $\overline{\tau_{xx}}$ by a value
proportional to the height of the terminus, $h_T$. This means that the terminal cliff exerts a nonlocal compressive force – the
higher the cliff, the stronger the compression. There is, of course, no physical basis for this compressive force, rendering the
Budd ice shelf irreparably problematic.

Perhaps even more problematically, from a pragmatic perspective, the deviatoric tension associated with Equation 31 is quite
a bit higher than that associated with the Thomas model. This is due to the factor of $\frac{1}{2}$, as opposed to the factor of $\frac{1}{4}$ in Equation
2. This can result in velocities substantially higher than those predicted by the Thomas model when thickness gradients are
steep (see Figure 2 for a concrete illustration of this, wherein the Budd model overestimates terminus velocities by about 40%).
The inclusion of vertical shear in the stress balance, therefore, is not just a technical point: failing to recognize the role of $\tau_{xz}$
can easily result in the construction of a bad ice shelf model.

## 7.2 An exact analytic ice shelf model

While we have shown that vertical shear stress must be included in a nonuniform-thickness ice shelf model, we have also shown
that the plug flow assumption precludes any well-defined vertical shear strain rate associated with that stress. To demonstrate
what such a vertical shear strain rate would look like, we now provide an alternative formulation of the Thomas model, which
does permit vertical differences in horizontal velocity.

In our construction, we alter the typical ice shelf assumptions ever-so-slightly, considering $\tau_{xx}$, rather that $u_x$, to be invariant
with depth. This leaves open the possibility that $u_x$ and $\dot\varepsilon_{xx}$ vary with $z$. We begin by considering what description of $\tau_{xz}$
would permit a solution to the $x$-momentum equation under this setup. For convenience, we rewrite the $x$-momentum equation
below.

$$2\frac{\partial}{\partial x}\tau_{xx} + \frac{\partial}{\partial z}\tau_{xz} = \rho_i g h \tag{32}$$

To satisfy Equation 32, $\tau_{xz}$ cannot be uniform in $z$. This follows from the use of the basal and surface boundary conditions
given by setting $\tau_{sx} = \tau_{bx} = 0$ in Equations 18 and 19: the same value of $\tau_{xz}$ cannot satisfy both unless the shelf is of uniform
thickness, and, in that case, the appropriate value of $\tau_{xz}$ is zero. Therefore, $\tau_{xz}$ must depend on $z$. Moreover, this dependence
must be linear: by our own stated assumptions, neither $2\frac{\partial}{\partial x}\tau_{xx}$ nor $\rho_i g h$ depend on $z$, and so a solution to this system precludes
$\frac{\partial}{\partial z}\tau_{xz}$ depending on $z$. The unique linear description of $\tau_{xz}$ preserving both the surface and basal boundary conditions is

$$\tau_{xz} = \frac{2z}{h}\tau_{xx}\frac{\partial}{\partial x}h. \tag{33}$$

This is our analytic description of the vertical shear stress in an ice shelf. To verify that this is a reasonable description, we
insert it into the $x$-momentum equation, obtaining the governing equation



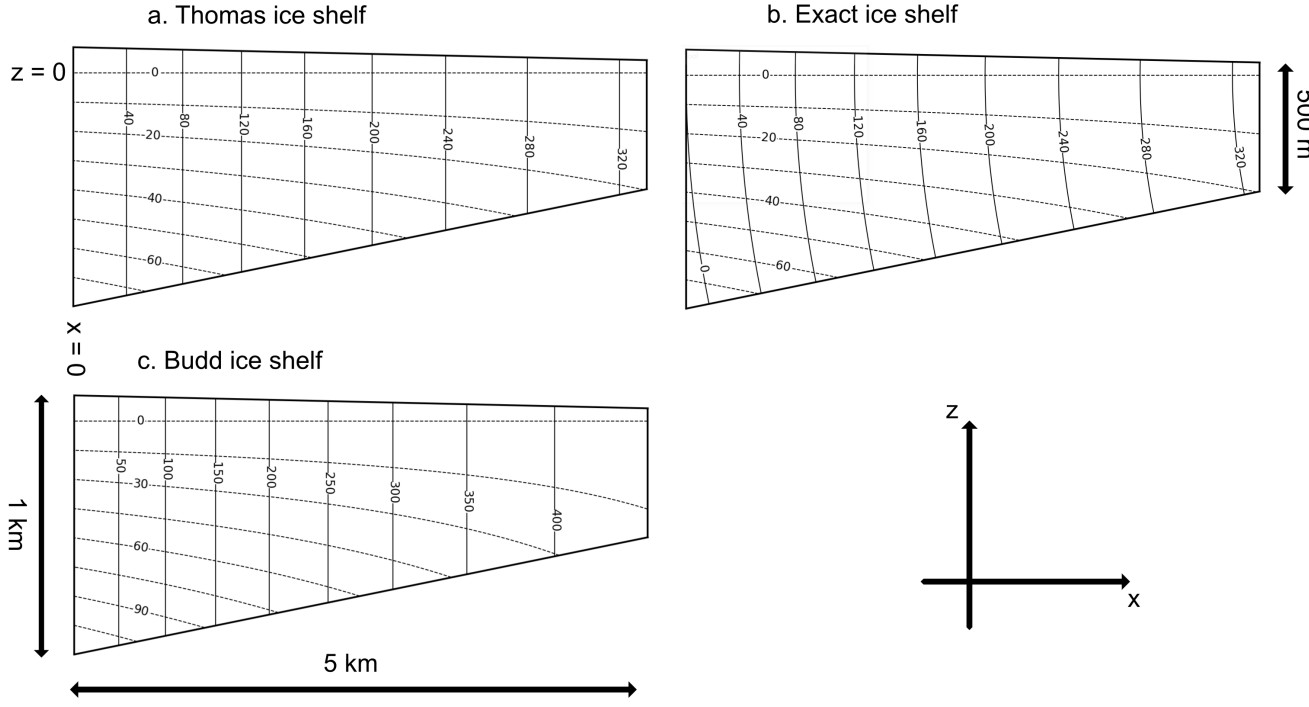

**Figure 2.** A comparison of three distinct ice shelf models, each of which treats vertical shear differently. Solid (approximately vertical) lines are contours for $u_x$, and dashed (approximately horizontal) lines are contours for $u_z$. Velocities are in m yr$^{-1}$ and are relative to the velocity at $(0, 0)$ (i.e., the upstream boundary at the waterline). The thickness of each shelf varies linearly from $H = 1$ km at the upstream boundary to $H = 500$ m at the terminal cliff, which is 5 km downstream. a) The conventional Thomas shelf, which includes vertical shear in the stress balance but requires plug flow. b) The exact velocity solution provided by Equation 36, in which vertical shear is both included in the stress balance and supported by the velocity field. c) The problematic Budd shelf of Equation 30, in which vertical shear is neglected from the stress balance. In each of the panels, we have used $\rho_i = 917$ kg m$^{-3}$, $\rho_w = 1018$ kg m$^{-3}$, and we have chosen the flow parameter $A$ to be consistent with the viscosity value of Jarosch (2008) under linear rheology.





$$2\frac{\partial}{\partial x}\tau_{xx} + \frac{2}{h}\tau_{xx}\frac{\partial}{\partial x}h = \rho_i g\frac{\partial}{\partial x}h. \tag{34}$$

Equation 34 is a first order linear ordinary differential equation in $\tau_{xx}$, and a general solution exists for equations of this form. It can be verified by direct substitution that this solution is

$$\tau_{xx} = \frac{1}{4}\rho_i gh. \tag{35}$$

Equation 35, though not depth-averaged, is of the same form as the familiar Thomas solution, and we argue that its emergence here supports the plausibility of Equation 33 as a description of vertical shear.

Next, we derive an exact, analytic velocity solution for the specific case of an isothermal ice ramp (that is, an ice shelf with linearly decreasing thickness) with linear rheology (i.e., with $n = 1$ in the constitutive relation, so that strain rates are independent of the effective stress). Notice that the conflicts identified in Section 6 were independent of rheology, and so these simplifications makes the mathematics more tractable while still leaving all previous discussion fully relevant.

With the flow parameter $A$ uniform, and some constant boundary value $u_0$, it can be directly verified that the velocity description

$$u_x = \frac{1}{4}A\rho_i g\int_0^x h\,dx + \frac{5}{8}A\rho_i gz^2\frac{\partial}{\partial x}h + u_0 \tag{36a}$$

$$u_z = -\frac{1}{4}\rho_i ghz \tag{36b}$$

is consistent with the stress field given by Equations 33 and 35. For example, evaluating $\dot{\varepsilon}_{xz} = \frac{1}{2}\frac{\partial}{\partial x}u_z + \frac{1}{2}\frac{\partial}{\partial z}u_x$ and then applying the constitutive relation of Equation 23 with $n = 1$, the resulting value of $\tau_{xz}$ is exactly Equation 33 (notice that, since $h$ is a linear in $x$, the integral term can be evaluated analytically, and $\frac{\partial}{\partial x}h$ is a constant). In comparison, in an identical setting, the isothermal, linear Thomas solution would yield

$$u_x = \frac{1}{4}A\rho_i g\int_0^x h\,dx + u_0 \tag{37a}$$

$$u_z = -\frac{1}{4}\rho_i ghz. \tag{37b}$$

These solutions differ only in the inclusion $z^2$ term in Equation 36a, and, therefore, they exactly agree at the waterline, where $z = 0$ (or, in a uniform-thickness shelf, the $z^2$ term would vanish, and the two expressions would become equivalent). Because $\frac{\partial}{\partial x}h \leq 0$, the exact solution for $u_x$ is less than the Thomas solution whenever $z \neq 0$. The largest disagreement between the two will occur at the base of the shelf, where $z$ attains its largest magnitude (see Figure 2), and the magnitude of the discrepancy



scales with the surface elevation gradient. At any depth, the discrepancy between the Thomas solution and the exact solution (i.e., the solution under which the base moves more slowly than the surface) is likely to be too small to measure directly (Sanderson and Doake, 1979).

## 7.3 A note on the correct calculation of vertical shear stress in the SSA

In most applications of the SSA, the calculation of the vertical shear stress is not a priority (by construction, the primary aim of the SSA is to approximate the leading-order terms, which are normally interpreted to be horizontal velocities and membrane stresses). However, in some applications, it may be necessary to evaluate the vertical shear stress. For example, one area in which vertical shear may be of interest is the topic of iceberg calving, wherein glacier ice experiences brittle failure. In ice shelf settings, such failure often results in massive, rifted icebergs which can be tens of kilometers in length. Although tensile failure is typically treated as the primary mechanism of brittle fracture in glaciers (Colgan et al., 2016), a more complete description of iceberg calving would include the possibility of shear failure as well (for example, see Bassis and Walker (2012)). Given an appropriate calculation of the vertical shear stress, a shear failure mechanism might be meaningfully discussed by even shallow approximations.

A natural approach to calculating the vertical shear stress would be to obtain a solution for the vertical velocity, use the plug flow property to obtain $\dot{\varepsilon}_{xz} = \frac{1}{2}(0) + \frac{1}{2}\frac{\partial}{\partial x}u_z$, and then apply the constitutive relation to obtain $\tau_{xz}$. However, this procedure is guaranteed to yield a physically unrealistic value for the vertical shear stress. By the workflow in Section 6, this approach will provide a description of $\dot{\varepsilon}_{xz}$ which is negative below the waterline (see Equation 26). Therefore, the corresponding expression for $\tau_{xz}$ will be negative below the waterline. As indicated by our analytic description of the vertical shear stress (Equation 33), the vertical shear stress should be *positive* beneath the waterline. Consequently, this approach to estimating $\tau_{xz}$ is inappropriate in shallow shelf settings.

An alternative approach is to vertically integrate the $x$-momentum equation. If this is done from the waterline to arbitrary elevation $z$, integration of Equation 6 yields

$$\tau_{xz} = \rho_i g z \frac{\partial}{\partial x}h - 2\int\limits_0^z \frac{\partial}{\partial x}\tau_{xx}dz, \tag{38}$$

where we have used the condition that $\tau_{xz}|_{z=0} = 0$ (see the footnote in the discussion preceding Equation 26). In the simplifying isothermal case, for example, and using the conventional description $\overline{\tau_{xx}} = \tau_{xx} = \frac{1}{4}\rho_i gh$, this is satisfied by $\tau_{xz}$ as written in our analytic expression, Equation 33. This approach, therefore, provides values in line with expectations.

For situations in which the magnitude of $\tau_{xz}$ is of interest, we recommend that vertical shear from the SSA be calculated via integration of the momentum equations, and we advise against calculating the term directly, via the velocity gradients emerging as SSA solutions. For back-of-the-envelope calculations, our analytic description of vertical shear stress, Equation 33, may be used.





## 8 Concluding remarks

In discussions of shallow ice shelf models, it is fairly common to hear vertical shear spoken of as "zero," "neglected," or otherwise unimportant. However, except in the simplest, uniform-thickness analytic models, this cannot be the case. Fundamentally, thickness gradients in ice shelves need to be balanced by vertical shear stress (Sanderson and Doake, 1979), and vertical shear stress should be neglected only to the extent that it is desirable to neglect a thickness gradient. While this is known, it is often forgotten, or, at least, discussed with potentially misleading imprecision. Failing to note the relationship between vertical shear and thickness gradients puts us at risk of constructing ice shelf models with exaggerated rates of extension and implausible ice front dynamics (see Figure 2 in Section 7.2). However, because shallow shelf models typically approximate horizontal velocities as depth-invariant, the velocity fields predicted by these models are not strictly consistent with nonzero vertical shear stress. Evaluation of vertical shear in ice shelves, when necessary, must be done via integration of the momentum equations, rather than via the velocity field directly.

*Author contributions.* CM devised the project and wrote the manuscript, with guidance from TB and EE. TB and EE both reviewed and edited, and acquired the financial support necessary for the project. TB directly supervised CM.

*Competing interests.* The authors declare that they have no conflict of interest.

*Acknowledgements.* This work was made possible by grants 1716865 and 1933105 from the U.S. National Science Foundation, and grant 80NSSC18K1477 from NASA.



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
