# Peer review of "Revisiting the role of vertical shear in analytic ice shelf models"

_EGUsphere, 2022_

## Referee Comment (RC4)

**Referee report for "Revisiting the role of vertical shear in analytic ice shelf models" by Miele et al.**

In this work, Miele et al. attempt to clarify any confusion that might have arises from the inclusion (or lack thereof) of vertical shear in the equations describing ice shelf flow, and how it relates to the derivation of an expression for depth averaged along-flow deviatoric stress, $\bar{\tau}_{xx} = \rho g h /4$, which is commonly invoked in the literature. They provide a historical perspective, describing different studies which have derived this relationship and the assumptions underpinning such. In particular, they highlight that complete ignorance of vertical shear stress is incompatible with fully neglecting surface slopes. They go on to describe a construction of the vertical shear stress in an ice shelf, which is offered as a way to determine such stresses in situations where they are required.

I found this paper somewhat tricky to review, not least because I don't think this is necessarily a 'scientific paper' in the conventional sense: the main aim of the paper is to clarify misconceptions that might arise on the construction of models in the past. Furthermore, I do not think that the 'new' part of the paper (the construction of the shear stress in S7) is indeed new (see below). I think this paper could be useful, particularly to students or those new to the field, but I am unsure whether it is a research paper, per se.

A note on framing: the authors state that "many authors still interpret vertical shear as absent in contemporary ice shelf analysis" and then list many mentions of similar language. As far as I see it, these mentioned authors are saying that the vertical shear stress term is not included at leading order, i.e. that $\partial u_x / \partial z = 0$, which is certainly true to leading order in the aspect ratio (in fact, to order (aspect ratio)^2 — see below written notes, particularly equation 15 therein). The distinction between neglected (in an asymptotic sense) and ignored (i.e. removed from the equations completely) is clear to these mentioned authors, I am sure. However, I am not so sure that this distinction is clear to students and, possibly, those unfamiliar with asymptotic analysis, and thus therein lies the niche of this paper.

More on the asymptotic analysis: this case has been described previously in rigorous detail by Schoof and Hindmarsh (10.1093/qjmam/hbp025, see their "S3.4: Fast Sliding (ii)"). The paper of Schoof and Hindmarsh is fairly intense; below, I have expressed their work in the notation of the present work. In particular, they show that $\partial u_x / \partial z = 0$ at leading order and, although they do not derive it explicitly, it is only a small step from their analysis to the linear stress term of S7.1 of the present paper. Importantly, they do not assume that $\partial \tau_{xx} / \partial z = 0$ (as is assumed by the present paper), but rather show that it emerges at leading order from the Euler equations, i.e. the assumption made in S7 is not necessary. I believe the present paper could be useful in translating this into more digestible language, but the authors should be clear this is not original.

A further point: this analysis shows that $\partial u_x / \partial z = 0$, as assumed by the Thomas model. The authors then go on to show that this leads to a contradiction; however, in the formal asymptotic framework, this is not a contradiction: the terms in their equation 29 are lower order and would be balanced by lower order corrections in the stresses.

Finally, I would say this this paper was quite difficult to read. I offer several suggestions to improve the readability of this paper: (1) many equations are referenced by number a long way from where they are expressed in the text. I would suggest giving them names to prevent having to flick back and forth (e.g. the x momentum equation (6) shows…), (2) I wonder whether it would be clearer to simply explain in words (assuming you do not want to include the rigorous analysis) the different assumptions, and then add derivations in appendices, (3) a table with different models, their assumptions, their expression for deviatoric stress, etc would help the reader to distinguish the models.

$$P + \tau_{xx} = (h - z) + O(\varepsilon^2) \tag{1}$$

$$2\frac{\partial \tau_{xx}}{\partial x} + \frac{\partial \tau_{xz}}{\partial z} - \frac{\partial h}{\partial x} = O(\varepsilon^2) \tag{2}$$

$$\left[\varepsilon = \frac{[z]}{[x]} \cdot \text{aspect ratio}\right]$$

$$\frac{\partial u_x}{\partial x} + \frac{\partial u_z}{\partial z} = 0 \tag{3}$$

$$\frac{\partial u_x}{\partial x} = \left(\tau_{xx}^2 + 2\varepsilon^2 \tau_{xz}\right)^{\frac{n-1}{2}} \tau_{zx} \tag{4}$$

$$\frac{\partial u_x}{\partial z} = 2\varepsilon^2 \left(\tau_{xx} + \varepsilon^2 \tau_{xz}\right)^{\frac{n-1}{2}} \tau_{xz} + O(\varepsilon^2) \tag{5}$$

Pose asymptotic expansion:

$$P = P_0 + \varepsilon^2 P_2 + \dots \tag{6}$$

$$\tau_{xx} = \tau_{xx}^0 + \varepsilon^2 \tau_{xx}^2 + \dots \tag{7}$$

$$\tau_{xz} = \tau_{xz}^0 + \varepsilon^2 \tau_{xz}^2 + \dots \tag{8}$$

$$u_x = u_x^0 + \varepsilon^2 u_x^2 + \dots \tag{9}$$

$$u_z = u_z^0 + \varepsilon^2 u_z^2 + \dots \tag{10}$$

Inserting (6) - (10) into (1) - (5) and retaining only leading order terms, we obtain:

$$P_0 + \tau_{xx}^0 = h - z \tag{11}$$

$$2\frac{\partial \tau_{xx}^0}{\partial x} + \frac{\partial \tau_{xz}^0}{\partial z} = \frac{\partial h}{\partial x} \tag{12}$$

$$\frac{\partial u_x^0}{\partial x} + \frac{\partial u_z^0}{\partial z} = 0 \tag{13}$$

$$\frac{\partial u_x^0}{\partial x} = \left(\tau_{xx}^0\right)^{n-1} \cdot \tau_{xx}^0 \tag{14}$$

$$\frac{\partial u_x^0}{\partial z} = 0 \tag{15}$$

(15) implies that $u_x^0 = u_x^0(x)$ ie the horizontal velocity is independent of depth to second order in the aspect ratio!

Then $\frac{\partial u_x^0}{\partial x}$ is independent of $z$, so, by (14), so is $\tau_{xx}^0$, ie $\tau_{xx}^0$ is independent of $z$, to leading order in $\varepsilon^2$.

Now we're at the stage reached by Miele et al, after they assume $\tau_{xx}$ independent of $z$.

(Aside: I think the linearity of $\tau_{xz}^0$ can more easily be seen by then taking $\frac{\partial}{\partial z}$ of (12):

$$2\underbrace{\frac{\partial^2 \tau_{xx}^0}{\partial x \partial z}}_{=0 \text{ because } \frac{\partial \tau_{xx}^0}{\partial z} = 0} + \frac{\partial^2 \tau_{xz}^0}{\partial z^2} = \underbrace{\frac{\partial^2 h}{\partial x \partial z}}_{=0 \text{ because } h = h(x)}$$

ie $\quad \frac{\partial^2 \tau_{xz}^0}{\partial z^2} = 0 \tag{*}$, so $\tau_{xz}$ is linear in $z$.

Then the solution to (*) satisfying BCs is $\tau_{xz}^0 = \frac{2z}{h}\tau_{zx}^0\frac{\partial h}{\partial x}$, where $\tau_{xx}^0$ must be found by solving other equations...)

See eqn (3.72) in Schoof + Hindmarsh (2010).

---

## Author Response (AR1)

We thank the editors and reviewers for their detailed and constructive engagement with this manuscript. At the suggestion of the Editor, we have reframed and significantly shortened the paper into a Brief Communication.

Original reviewer and editor comments are in blue, sans serif, 10 pt. Helvetica font.
Responses to reviews are in black, serif, 12 pt. Times New Roman font.

Reviewer 1

General comments:

I think that I reviewed this manuscript for another journal, and if so, I was supportive of its publication at that time, and still am.  I do note some relatively minor, but important corrections to the "narrative" being presented, and I describe them now:

The abstract uses the words "miscommunications", "misinterpretation" and "misconception".  I think that these words are sort of unfair to the early scientists who developed the initial modes of thinking about, doing analysis with and modeling ice shelves.  These early scientists were well aware of the fact that shear stress in the vertical was prevalent in ice shelves at amplitudes that could be large (e.g., at an ice front or when there are large thickness gradients); however, their intention was to develop strategic simplifications and approximations which would allow glaciological science to make progress.  Their pioneering work leading to the "shallow shelf approximation" was fundamental to the progress of glaciology through the 1960's onward to the present day.  It is thus not only unfair to their legacy to imply that they were "misleading", but it is a kind of cheap writer's trick to introduce the substance of the present paper.  I strongly object to this tone and think that it detracts from the paper by setting up a false "combative" tone that completely misleads the reader.

I see that this tone that I object to is not present in the Introduction, and the authors very correctly laud the initial development of one of the most effective approximations in glaciological history (the shallow shelf approximation).  This is important. And I compliment the authors for having done so.  But again: I see words like (line 65) "persistent mischaracterizations".  This is a false and incorrect statement: approximation is not a mischaracterization.

Again, thank you for pointing out that the tone has been, at times, at odds with our intention. To reiterate from our online response, our intention is to identify, and resolve, a perceived mismatch between how modelers talk about vs. how they actually construct modern ice shelf models today. We have adjusted our sentence structure to better describe this present-day miscommunication, without incorrectly implicating the pioneering authors discussed. For example, one line you have mentioned (previously line 65, now line 20) now reads: "...a topic which is sometimes incompletely communicated *today*."

The revised text also acknowledges that any miscommunication is, at least partially, the responsibility of the hypothetical reader. For example, beginning on line 46: "it is common to encounter language which, *to a novice glaciologist, might seem to imply* that vertical shear is still discarded entirely." As we have mentioned in the online response, we assume that the language we identify in recent papers is mostly shorthand for a more nuanced understanding on the part of the authors (as also suggested by Reviewer 4). However, because this shorthand is so pervasive in modern literature, our opinion is that a novice glaciologist would never encounter any reason to suspect deeper nuance – hence, the possibility of miscommunication.

A challenge:  After recently attending the AGU and also reading a paper by Catherine Walker:

Walker, C. C. and Gardner, A. S. (2019). Evolution of ice shelf rifts: Implications for formation mechanics and morphological controls, Earth and Planetary Science Letters, 526,115764, doi:10.1016/j.epsl.2019.115764.

I became aware of the fact that many rifts on the Antarctic ice shelves are not vertical, but slightly offset from vertical, and that they have an interesting, not-fully-understood asymmetry of the rift shoulders associated with bending moments.  I wonder if this phenomena (also described in one of Walker's papers on ice shelled planets) is an observable phenomena that is directly related to the subject of this paper.  If the authors think that it is, then they might find that their paper is made even stronger by including references to the Walker study, and also to (not sure if this is as relevant):

Walker, C. C., J. N. Bassis, J. N. and Schmidt, B. E. (2021). Propagation of vertical fractures through planetary ice shells: The role of basal fractures at the ice-ocean interface and proximal cracks. The Planetary Science Journal, Vol. 2, No. 4, doi:10.3847/PSJ/ac01ee.

Unfortunately, due to length constraints of the updated Brief Communication format, we weren't able to incorporate these topics.

Specific comments:

line 2 of abstract "…, extending in only one direction,…". I'm pretty sure that Weertman's 1957 paper also gives the solution for spreading in two horizontal directions.

This is a good point. We've tweaked our description so that, although we still only discuss Weertman's uniaxial solution, our wording avoids the incorrect implication that uniaxial extension was the *only* case solved by Weertman. See lines 7 – 9.

I was not able to find other errors or edits to make, and I commend the authors for doing a fine job of proof reading

Reviewer 2

The manuscript is concerned with the mathematical modeling of ice shelf flow. In particular, the authors focus on the role of vertical shear in thin-film approximations of the ice flow problem that are valid for the case of negligible basal friction. The authors state that their goal is to clarify what is in their view a misunderstanding in the main stream glaciological literature, i.e., that "it remains common to misinterpret vertical shear stress as typically neglected in current ice shelf modeling studies".

As much as I understand that the intention behind this manuscript is constructive, and as much as I share the authors' opinion that the published literature is somewhat confusing, I am afraid I have to express a negative opinion about the content and originality of this work. In fact, in my view this problem has been solved already in rigorous mathematical terms in publications concerning the asymptotic structure of the Stokes problem in the limits of thin flows (low aspect ratio) and extensional stresses much larger than the vertical shear stresses.

We largely agree with the reviewer here. Specifically, we concur that a conventional treatment of vertical shear has been conclusively established by other authors – indeed, this is the basis for our pointing out that the treatment of vertical shear has changed over time. Additionally, we all seem to agree that a fair bit of published literature is somewhat confusing, to the extent that it appears, at times, to misrepresent the established treatment of vertical shear. We have rewritten this paper as a Brief Communication, which we hope the reviewer will agree is a better format for this clarification.

Two main strands of research support my statement: Doug MacAyeal' s seminal shelfy-stream paper (JGR 1989, **Large-scale ice flow over a viscous basal sediment: theory and application to Ice Stream B, Antarctica,** see model derivation in appendix A), and more recently Schoof and Hindmarsh, 2010 (**Thin-Film Flows with Wall Slip: An Asymptotic Analysis of Higher Order Glacier Flow Models** https://academic.oup.com/qjmam/article/63/1/73/1843730, esp. sec. 3.4, the limit of lambda << epsilon). For completeness, and purely as a side note, the only difference between these two derivations being that Schoof and Hindmarsh retain the stress ratio lambda <<1 (see eq. 2.18, lambda is the ratio between shear stress and extensional stress, whereas epsilon<<1 is the aspect ratio, both taken to be small; then lambda<<1 describe the case of ice flow dominated by extensional stresses, as is the case in an ice shelf), whereas MacAyeal assumes the distinguished limit lambda ~ epsilon^2 with (lambda, epsilon) <<1 (or, in MacAyeal's notation, epsilon ~ delta^2), which is one peculiar case of the broader model category obtained in Schoof and Hindmarsh's more general limit of lambda << epsilon. Yet, despite this minor formal difference, they obtain the same leading-order model, as noted by Schoof and Hindmarsh.

To illustrate my point, I will refer to MacAyeal's derivation, which is simple and clear; in his derivation, e_{xz}, e_{yz} are the vertical shear stresses therein. I now use epsilon to mean stress ratio, as per MacAyeal's notation.

The first key point is about the zeroth-order momentum balance: Yes, to leading order the solution reads e_{xz}^{0} = 0, e_{yz}^{0}=0 (eqs. A25 and A26), but all that means, as the derivation clearly explains, is that these stresses are of order epsilon (epsilon being the aspect ratio) compared to the extensional stresses and the lateral shear stress. This is very different from stating that the vertical shear stresses are zero - rather, it only says that extensional stresses are small compared to extensional and lateral shear stresses.

To find the leading order expressions for the vertical shear stresses, one needs to carry on with the expansions to the next (first) order. As expected, the order epsilon corrections e_{xz}^{1}, e_{yz}^{1} appear in the first order (order epsilon) momentum conservation equations (A30 - A31), as they do in the

dynamic boundary conditions that determine the horizontal gradients of the ice shelf surface and bottom (A34-A35).

The misconception pointed out by the authors arises, if I understand correctly, by the fact that thanks to careful algebraic manipulations $e_{xz}^{1}$, $e_{yz}^{1}$ disappear from the final momentum conservation equations (A 36 and A37) that must be solved to determine $e_{xx}^{0}$, $e_{yy}^{0}$, $e_{xy}^{0}$. Yet, that does not mean that vertical shear stresses are zero - in fact, they can be computed diagnostically by vertical integration of the constitutive relation with appropriate boundary conditions, once the problem for $e_{xx}^{0}$, $e_{yy}^{0}$, $e_{xy}^{0}$ has been solved.

This topic fits well with the comments of Reviewers 3 and 4, who also brought up the dimensional analysis basis for the SSA's construction. We have included a brief commentary on dimensional analysis and its application to the SSA's vertical shear. See the paragraph beginning on Line 35, as well as Section 3. In these sections, we primarily reference Weis et al. (1999) simply because their equations are more directly applicable to our discussion, and we did not have the space to perform long derivations or include detailed analyses. However, we do cite both of the papers identified above (see line 32, for example).

It is my view that this work, published over 30 years ago, already clarifies the issue raised by the authors without any possible doubt, while Schoof and Hindmarsh 2010 provide a further generalization. In light of this, it is my opinion that the manuscript as is lacks the level of originality that is required to warrant publication.

Given that the primary concern is one of originality, we hope that the reviewer will consider this topic appropriate as a Brief Communication.

Reviewer 3

This paper is aiming to clarify to the broader glaociology community some assumptions made in models for ice shelfs, specifically the relation between vertical shearing and surface gradients. I think such a topic/disucssion is welcome, but the presentation of this needs to be improved in order to reach the inteded audience. I suggest the following edits:

1) Formalize the arguments using perturbation expansions and scalings. The vertical shearing is not "absent", it is just of a higher order in the perturbation expansion I presume (i.e. "neglected"). Is it of the same order as the surface gradients perhaps? I recommend connecting your arguments to the analysis by Schoof and Hindmarch (mentioned here by another reviewer). A large part of the glaciological community is familiar with perturbation expansions and scalings, and for the ones who are not, they could learn it from this paper. As the papers intent seems to be educate this should fit in well. You don't need to erase your other arguments, but I recommend adding perturbation expansions as an supporting, more formal, explanation.

We have included a brief commentary on dimensional analysis and its application to the SSA's vertical shear stress. See the paragraph beginning on Line 35, as well as Section 3. In these sections, we primarily reference Weis et al. (1999), rather than the suggested paper, simply because their equations are more directly applicable to our discussion, and, given the reduced format, we did not have the space to go beyond the absolute basics.

2) The exeriment in Figure 2 is nice but in my opinion it could be made more interesting. For the purpose of showing that this is actually important in real ice shelf simulaitons, I would extend this experiment, make it more realstic - maybe some variation in the ice surface.

We had to cut this discussion due to space constraints.

3) Overall it is sometimes a bit unnecessary difficult to follow the derivations, think about if there is a clearer way to present it. Make the two derivations side by side perhaps?

This issue was also brought up by Reviewer 4. We believe this issue is solved by the shorter format.

4) As mentioned by another reviewer, make sure that its not possible to percieve the tone in a bad way, although I am sure this is not the intent.

We've been more cautious about our language in this draft. Please see our response to Reviewer 1 on the first page of this document.

Reviewer 4

In this work, Miele et al. attempt to clarify any confusion that might have arises from the inclusion (or lack thereof) of vertical shear in the equations describing ice shelf flow, and how it relates to the derivation of an expression for depth averaged along-flow deviatoric stress, $\tau_{xx} = \frac{1}{4}\rho gh$, which is commonly invoked in the literature. They provide a historical perspective, describing different studies which have derived this relationship and the assumptions underpinning such. In particular, they highlight that complete ignorance of vertical shear stress is incompatible with fully neglecting surface slopes. They go on to describe a construction of the vertical shear stress in an ice shelf, which is offered as a way to determine such stresses in situations where they are required.

I found this paper somewhat tricky to review, not least because I don't think this is necessarily a 'scientific paper' in the conventional sense: the main aim of the paper is to clarify misconceptions that might arise on the construction of models in the past. Furthermore, I do not think that the 'new' part of the paper (the construction of the shear stress in S7) is indeed new (see below). I think this paper could be useful, particularly to students or those new to the field, but I am unsure whether it is a research paper, per se.

We have rewritten this paper as a Brief Communication, which we hope the reviewer will agree is a more appropriate format for this topic.

A note on framing: the authors state that "many authors still interpret vertical shear as absent in contemporary ice shelf analysis" and then list many mentions of similar language. As far as I see it, these mentioned authors are saying that the vertical shear stress term is not included at leading order, i.e. that $\frac{\partial u_x}{\partial z} = 0$, which is certainly true to leading order in the aspect ratio (in fact, to order (aspect ratio)^2 — see below written notes, particularly equation 15 therein). The distinction between neglected (in an asymptotic sense) and ignored (i.e. removed from the equations completely) is clear to these mentioned authors, I am sure. However, I am not so sure that this distinction is clear to students and, possibly, those unfamiliar with asymptotic analysis, and thus therein lies the niche of this paper.

We do agree that the language used by these mentioned authors is likely shorthand for a more nuanced understanding – but it is our opinion that the nuance is not conveyed in a way which would be picked up by a reader not already familiar with the topic (please see our first response to Reviewer 1 on the first page of this document).

More on the asymptotic analysis: this case has been described previously in rigorous detail by Schoof and Hindmarsh (10.1093/qjmam/hbp025, see their "S3.4: Fast Sliding (ii)"). The paper of Schoof and Hindmarsh is fairly intense; below, I have expressed their work in the notation of the present work. In particular, they show that $\frac{\partial u_x}{\partial z} = 0$ at leading order and, although they do not derive it explicitly, it is only a small step from their analysis to the linear stress term of S7.1 of the present paper. Importantly, they do not assume that $\frac{\partial \tau_{xx}}{\partial z} = 0$ (as is assumed by the present paper), but rather show that it emerges at leading order from the Euler equations, i.e. the assumption made in S7 is not necessary. I believe the present paper could be useful in translating this into more digestible language, but the authors should be clear this is not original.

We have included a brief discussion of dimensional analysis in our revised manuscript. Unfortunately, due to space constraints of the new format, we found it necessary to keep this discussion very simplistic, and we primarily reference Weis et al. (1999) because their equations are more immediately applicable to our discussion without prohibitively long derivations. See the paragraph beginning on line 35, as well as Section 3. However, we do now credit Schoof and Hindmarsh with the linear shear stress solution (line 74).

A further point: this analysis shows that $\frac{\partial u_x}{\partial z}$, as assumed by the Thomas model. The authors then go on to show that this leads to a contradiction; however, in the formal asymptotic framework, this is not a contradiction: the terms in their equation 29 are lower order and would be balanced by lower order corrections in the stresses.

This section no longer appears in the revised manuscript.

Finally, I would say this this paper was quite difficult to read. I offer several suggestions to improve the readability of this paper: (1) many equations are referenced by number a long way from where they are expressed in the text. I would suggest giving them names to prevent having to flick back and forth (e.g. the x momentum equation (6) shows…), (2) I wonder whether it would be clearer to simply explain in words (assuming you do not want to include the rigorous analysis) the different assumptions, and then add derivations in appendices, (3) a table with different models, their assumptions, their expression for deviatoric stress, etc would help the reader to distinguish the models.

We believe the shortened format has removed these issues.

Editorial decision

Dear Authors and Reviewers,

thanks for a vivid discussions phase, which I have now ended after receiving four reviews. In order to consider publication in The Cryosphere, the manuscript requires substantial revisions likely including a change of the article type from "research article" to "brief communication".  My more detailed comments are as follows:

[1] I would like to thank all reviewers. Their comments demonstrate a strong expertise in the foundations of ice dynamic modelling including the asymptotic analysis required in modelling ice-shelf flow. Many constructive suggestions are provided, including explicit derivations linking earlier works to the results of this study.

[2] There is no consensus between reviewers as to whether or not this paper satisfies the criteria of a research article. The main concern synthesizing from the more critical reviews (RC2, RC4) is a lack of novelty, e.g., compared to earlier works such as Schoof and Hindmarsh (2010). Here, the authors agree that there is no "discovery on which we claim novelty." Instead they argue that "..all students of glaciology will know that negligible vertical shear is not a necessary component of a shallow shelf model". This point, however, is confronted by multiple comments (RC3 & RC4)  that the paper is hard to read even for some invited experts. It is therefore doubtful if this objective will be achieved.

[3] In summary, I believe that the lack of novelty precludes classification of this article as a "research article". Going forward, I suggest that the authors provide a significantly revised version that takes the reviewer's comments into account and delivers a clear message to the intended target audience in form of a "brief communication". This is difficult to do for such a topic, and it will require substantial shortening and reframing. I also understand that there might be different understandings for the terms "novelty" & "research articles etc." between different authors, reviewers, and editors. However, given the reviews and the publication criteria of The Cryosphere, I believe this will be the best option to go for.

Thank you again for all your work. Although this discussion contains some contraversial points, I surely appreciate this form of scientific exchange.

Kind regards,

Reinhard Drews

We appreciate your openness with regards to this project, and we have rewritten and reframed the manuscript into a Brief Communication, as suggested. As the only issues identified have been with regards to novelty (not enough) and complexity (too much), we believe the change in format to a four-page article will have adequately resolved these.

---

## Author Response (AR2)

We thank the editor and reviewer for their time and effort in helping to strengthen this manuscript.

Original reviewer comments are in blue, sans serif, 10 pt. Ariel font.

Responses to reviewer comments are in black, serif, 12 pt. Times New Roman font.

Since the purpose is to clarify some language/history I feel the language in the paper should be even clearer. There is still some confusion language around the words zero vs negligible. That some derivatives of vertical shearing is negligible in some equations does not mean it is considered zero. I think this is what the authors want to clarify but I think it can be made even clearer and explicit. Maybe do some definitions of these words.

We agree that clarity should be a primary goal here, and we appreciate having any potentially confusing language pointed out. We have emphasized, in the revised manuscript, the difference between a value being zero in reality vs. being neglected for the sake of constructing a useful approximation. For example, see our parenthetical clarifications in lines 38 and 65.

We do wish to note that this is a separate issue from the primary purpose of our manuscript. To reiterate our intention, we aim to point out that, *although historical modelers did neglect vertical shear, modern modelers do not* (as pointed out by the reviewer, and in our Equation 4b, modern modelers neglect only certain derivatives of vertical shear). In our readings of the literature, we have not personally noticed any confusion regarding the meanings of "negligible" and "zero," though we agree the distinction is important. We have slightly reworded the abstract to avoid misdirecting the reader toward that topic.

Also I would like to see 1) a plot of a computed solution to equations 6 (perhaps instead of the current figure),

Below, we have attached a plot illustrating the vertical shear stress solution of Equation 6b for a surface elevation gradient of $\frac{d}{dx}h = -0.001$. The vertical coordinate, z, spans from ~1000 meters below sea level to ~100 meters above sea level (representing a ~1 km thick shelf at flotation). This plot demonstrates the simple, linear dependence of vertical shear stress on z. Since Equation 6b, shown below, is such a simple relationship, and since 6a is already well established, we hope you might support our keeping the current figure instead of replacing it:

[Figure]

2) include some key paragraphs from the historical paper where they use the words "zero" and "negligible". I think this is important to do direct cites so that the historical papers are represented correctly.

We have provided some direct quotes, as requested. See line 23 for a quote by Thomas (1973) and line 28 for a quote by Sanderson and Doake (1979). Although we did not find the space to directly quote others, we refer the reviewer to Weertman's (1956) Equation 3, which states that "$\sigma_{xy} = \sigma_{xz} = \sigma_{yz} = 0$" and Robin's (1975) comment that they "assume the shear stresses in the xz and y planes to be zero."

In our interpretation, these quotations are intended only within the context of the authors' own approximate models (i.e., these quotations do not, to us, imply that the authors mistakenly believed vertical shear was exactly zero in real ice shelves). Rather, our takeaway is that these authors all neglected vertical shear, in contrast with modern authors.

Specific comments:

Page 1: state why we are interested in t_xx - intuitively one would be interested in e.g. velocity and surface elevation h

This is a good point. We now state, on line 11: "Using a depth-averaged constitutive relation, expressions of this form permit the calculation of strain rates and velocities."

Line 20-25: Write the mathematical symbol for vertical shear

Written (line 23).

Line 30. Is it really that they previously thought it is exactly zero? If so please cite some sentence from the papers where they say vertical shear is zero. Perhaps they thought that it was non-zero but negligible, only that they didn't formalise this thinking with perturbation expansions? Clarify.

We have provided some direct quotes, as requested (see one of our previous comments). Even so, we agree with the reviewer that it is unlikely these pioneering authors thought vertical shear was exactly zero in real ice shelves. In our interpretation, they describe vertical shear as being zero only in the context of their approximate models (i.e., they are saying that they have neglected the term). This is to be held in contrast with the modern approach, which neglects only certain derivatives of vertical shear, as you have noted above.